# Harnessing Brewery Spent Grain for Polyhydroxyalkanoate Production

**Robe D. Terfa** [1], **Priyanshi N. Patel** [1], **Hwidong D. Kim** [2], **Matthew D. Gacura** [3], **Gary Vanderlaan** [3], **Longyan Chen** [1], **Xiaoxu Ji** [1] and **Davide Piovesan** [1,*]

[1] Department of Biomedical Industrial and Systems Engineering, Gannon University, Erie, PA 16541, USA; terfa001@gannon.edu (R.D.T.); patel272@gannon.edu (P.N.P.); chen084@gannon.edu (L.C.); ji001@gannon.edu (X.J.)

[2] Department of Environmental Science and Engineering, Gannon University, Erie, PA 16541, USA; kim008@gannon.edu

[3] Department of Biology, Gannon University, Erie, PA 16541, USA; gacura001@gannon.edu (M.D.G.); vanderla002@gannon.edu (G.V.)

[*] Correspondence: piovesan001@gannon.edu (D.P.); Tel.: +1-814-871-7221

**Abstract:** The utility of brewery spent grain (BSG), a byproduct of the beer production process, for the synthesis of polyhydroxyalkanoates (PHAs), is a significant advancement towards sustainable and cost-effective biopolymer production. This paper reviews the upcycling potential of BSG as a substrate for PHA production, utilizing various biotechnological approaches to convert this abundant waste material into high-value biodegradable polymers. Through a comprehensive review of recent studies, we highlight the biochemical composition of BSG and its suitability for microbial fermentation processes. This research delves into different methodologies for PHA production from BSG, including the use of mixed microbial cultures (MMCs) for the synthesis of volatile fatty acids (VFAs), a critical precursor in PHA production, and solid-state fermentation (SSF) techniques. We also examine the optimization of process parameters such as pH, temperature, and microbial concentration through the application of the Doehlert design, revealing the intricate relationships between these factors and their impact on VFA profiles and PHA yields. Additionally, this paper discusses challenges and future perspectives for enhancing the efficiency and economic viability of PHA production from BSG. By harnessing the untapped potential of BSG, this research contributes to the development of a circular economy model, emphasizing waste valorization and the creation of sustainable alternatives to conventional plastics.

**Keywords:** solid-state fermentation; mixed microbial culture; volatile fatty acid; solid-state enzymatic hydrolysis

## 1. Introduction

Brewer's spent grain (BSG) is a promising substrate for the production of polyhydroxyalkanoates (PHAs), biodegradable polymers with diverse industrial applications. BSG, a common byproduct of beer production, is rich in organic compounds suitable for PHA synthesis [1]. Several studies have emphasized the potential of BSG in biotechnological applications, particularly in PHA production, due to its composition and abundance [2]. The process of utilizing BSG for generating PHAs involves cultivating microorganisms that extract nutrients from BSG, leading to the biological synthesis of PHAs via fermentation pathways [3]. Microorganisms play a crucial role in the biotransformation of BSG, aiding in the production of enzymes and metabolites of industrial interest, enhancing diets for humans and animals, and improving soil fertility [4]. While traditionally BSG has been utilized as animal feed due to its rich fiber and protein content, there is a growing interest in extracting various substances from BSG for further utilization [5,6]. Industrial applications of BSG not only leverage BSG as a valuable animal feedstock but also as a bioreactor input

for PHA synthesis [7]. Numerous studies have focused on optimizing microbial PHA production using BSG as a substrate, demonstrating the feasibility of utilizing this waste product for sustainable bioplastic production [8].

Polyhydroxyalkanoates (PHAs) are promising, sustainable alternatives to traditional plastics due to their biodegradability and biocompatibility, aligning with the principles of the circular economy and sustainable development. Polyhydroxyalkanoates (PHAs) and polyhydroxybutyrates (PHBs) are both biodegradable polymers produced by microbial fermentation, but they differ in their composition and properties. PHAs are a family of polyesters synthesized by numerous microbial taxa as intracellular carbon and energy storage compounds, typically under conditions of nutrient restriction. PHAs can vary widely in their monomeric composition, leading to polymers with different physical and mechanical properties. This variability intrinsically provides a certain amount of design cushion in assembled PHA products, permitting a suite of customizations, ranging from rigid thermoplastics to flexible elastomers.

PHBs, on the other hand, are the most common and well-studied members of the PHA family. They are polyesters composed entirely of 3-hydroxybutyrate units. PHBs are known for their high crystallinity and stiffness, which unfortunately results in brittleness at room temperature. This limits their applications when toughness and flexibility are required. However, their biodegradability and biocompatibility make them suitable for medical applications, such as sutures and drug delivery systems, as well as for packaging materials.

The primary difference between PHAs and PHBs lies in their polymer composition—PHAs represent a broad family with diverse molecular properties while PHBs are a specific PHA type with distinct characteristics. Generally, the flexibility in the monomeric composition of most PHAs makes them more versatile in applications than PHBs, which have more limited physical properties due to their homopolymeric nature.

The vast landscape of PHA monomers exceeds 150 varieties. Despite this diversity, 3-hydroxycarboxylates of 4–14 carbons, particularly 4-hydroxybutyrate (4HB) and 4-hydroxyvalerate (4HV), remain extensively researched due to their metabolization pathways and favorable polymer characteristics [9,10]. With a melting point ranging between 40 °C and 180 °C, exceptional resistance to hydrolytic attack, and degradation by UV light, PHAs exhibit non-toxicity, biocompatibility, and biodegradability, rendering them suitable for pharmaceutical and medical applications [9].

This paper delineates a comprehensive exploration into the valorization of BSG to produce polyhydroxyalkanoates (PHAs), highlighting two primary PHA production processes—mixed microbial cultures (MMCs) and solid-state fermentation (SSF)—and corresponding BSG pretreatment methodologies, including hydrolysis methods such as acid exposure, enzyme treatment, and plasma discharge. Through the meticulous examination of MMC and SSF strategies, along with optimized pretreatment techniques, we offer insight into maximizing the potential of BSG as a sustainable resource for biopolymer production. The strategic implementation of these processes not only underscores the versatility and efficiency of BSG in PHA production but also propels us toward a more sustainable and cost-effective approach to biopolymer manufacturing. By harnessing the inherent value of BSG and through the careful consideration of production and pretreatment methods, this study contributes significantly to the burgeoning field of waste-to-value, illustrating a promising pathway toward the advancement of bio-based materials and the development of a more circular economy.

## 2. Pretreatment Methods

Solid-state enzymatic hydrolysis (SSEH) and thermally diluted sulfuric acid hydrolysis are two distinct methods used in biomass conversion processes. The key difference between SSEH and thermally diluted sulfuric acid hydrolysis lies in the mechanisms of action and the agents used for biomass breakdown. SSEH relies on enzymatic action to catalyze the breakdown of complex biomass components into simpler sugars, while thermally diluted

sulfuric acid hydrolysis utilizes the acidic properties of sulfuric acid to hydrolyze the biomass components into fermentable sugars through a chemical process [11,12].

### 2.1. Thermally Diluted Sulfuric Acid Hydrolysis

Thermally diluted sulfuric acid hydrolysis involves the treatment of biomass with sulfuric acid under controlled temperature and time conditions to systematically break down the complex structure of the biomass and release sugars that can then be converted into volatile fatty acids (VFAs) via a subsequent microbial fermentation process (Figure 1). VFAs are organic chemicals with various applications, serving as carbon sources for microorganisms to produce useful metabolites that collectively act as starting materials for the synthesis of long-chain fatty acids and polyhydroxyalkanoates (PHAs) [13]. Thermally diluted sulfuric acid hydrolysis is effective in hydrolyzing hemicellulose and cellulose components of the biomass into fermentable sugars at the liquid state [12].

Acidic hydrolyses coupled to thermal treatment thus manipulate the microbial fermentation process to elevate VFA production from reclaimed, pretreated BSG. Prior research has examined the efficacy of BSG's transformation from solid waste to a valuable resource by producing VFAs via thermal diluted sulfuric acid hydrolysis followed by anaerobic fermentation, underlining the methodology pipeline's potential for PHA and PHB synthesis [1,8,10,14,15]. These research groups investigated overall metabolic pathways, reactor design, and biomass pretreatment methods for cost-effective PHA production. Untreated BSG can be difficult for microbes to ferment as a nutrient source whereas exposure to thermally diluted sulfuric acid hydrolysis extracts a carbohydrate-rich liquid fraction especially amenable for microbial metabolism. Following acid pretreatment, the liquid component of BSG is removed from the slurry to generate an acid hydrolysate [2]. To ensure purity, the acid hydrolysate is then sterile filtered and refrigerated before being employed in subsequent polyhydroxyalkanoate (PHA) production assays [2]. Anaerobic fermentation of this liquid media illustrated the scalability and feasibility of manufacturing volatile fatty acid (VFA) from BSG. Isolated bacterial species such as *Burkholderia cepacia*, *Bacillus cereus*, and *Cupriavidus necator* have been evaluated for BSG conversion into polyhydroxyalkanoates (PHAs), with promising PHA yields and accumulation test results confirming BSG's potential for value-added product production [9].

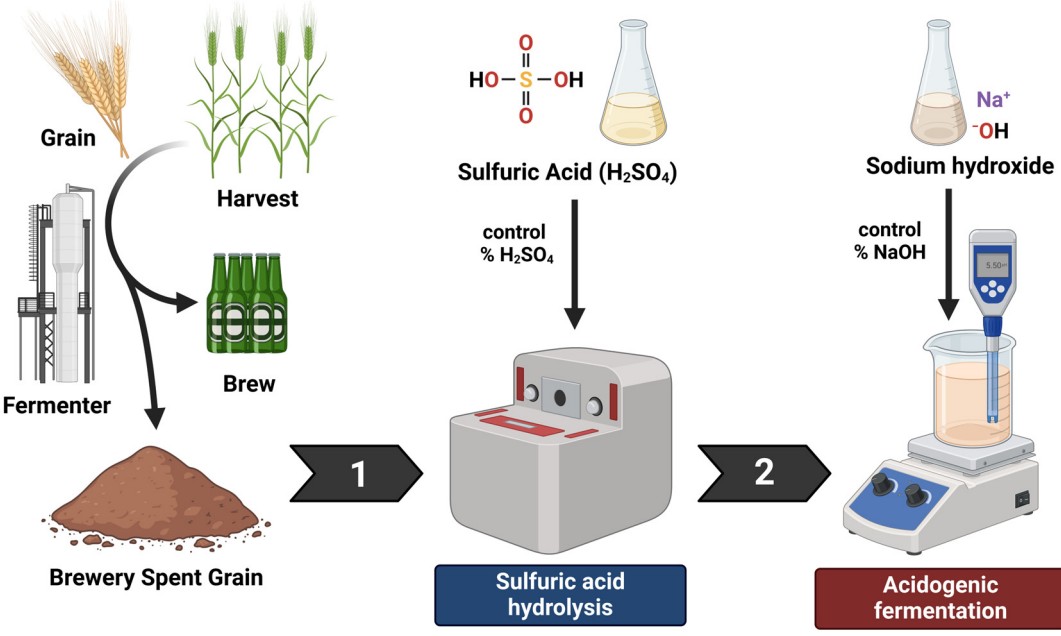

**Figure 1.** Schema for volatile fatty acid production from BSG using sulfuric acid hydrolysis and acidogenic fermentation leveraging anaerobic cultures. Adapted from [16]. Created in BioRender.

## 2.2. Solid-State Enzymatic Hydrolysis (SSEH)

Solid-state enzymatic hydrolysis (SSEH) involves the enzymatic breakdown of complex substrates, such as lignocellulosic biomass, into simpler components without the presence of free-flowing water (Figure 2). Typically, SSEH is performed in a solid-state environment, and, as a process, SSEH aims to release fermentable sugars from the normally recalcitrant biomass for subsequent metabolic pathways, including PHA production [11]. Additionally, research indicates that increasing the enzyme-to-substrate ratio can positively enhance the generation of bioactive peptides from BSG hydrolysates, and presumably, these generated oligopeptides might harbor nutritional potential for PHA-synthesizing microbial taxa [17]. A variety of source hydrolysates were comparatively examined, including BSG, grape pomace, and olive mill solid waste, which were each pre-treated using SSEH approaches before further pushed through an SSF production pipeline to drive the conversion of substrates into desired products through microbial activity [18,19].

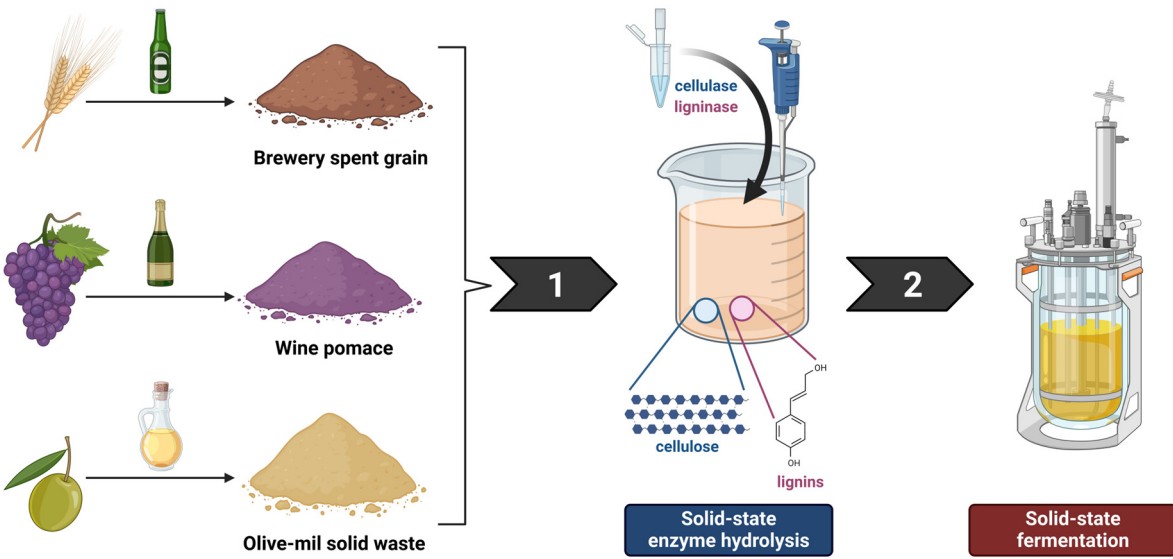

**Figure 2.** Schema of solid-state enzymatic hydrolysis (SSEH) and solid-state fermentation adapted from [18]. Created in BioRender.

SSEH focuses on the enzymatic breakdown of complex substrates into simpler compounds, primarily aiming to release fermentable sugars for subsequent bioprocesses [20]. Due to a central reliance on enzyme kinetics, SSEH efficacy is dictated by factors that govern any given organism's specific enzyme function, namely catalytic characteristics such as optimal temperature, pH, osmotic pressure, and type of enzyme extract used [20].

## 2.3. Catalysis and Non-Ionizing Radiation

Metal catalysts have also been evaluated for productive VFA yields, in particular for the generation of carboxylic acids [13]. Trace metal supplementation has been identified as an additional method to boost VFA utilization [21]. Bioreactor studies have shown that metals like copper (Cu) and iron (Fe) in anaerobic digestion processes can greatly impact acidogenic biomass, potentially leading to an increase in VFA concentration [22]. Furthermore, nanoparticles of metal substances such as iron (Fe) and nickel (Ni) have been effectively impregnated into char support, improving thermal stability with encouraging increases in VFA production [23].

Enhancing the enzymatic hydrolysis of BSG with nonthermal plasma offers a combinatorial approach to improving the valorization of recalcitrant substrates (Figure 3). Recent investigations have explored the use of nonthermal plasma to optimize the enzymatic hydrolysis of BSG, providing a promising method to enhance the conversion of BSG components into useful biomaterials [24].

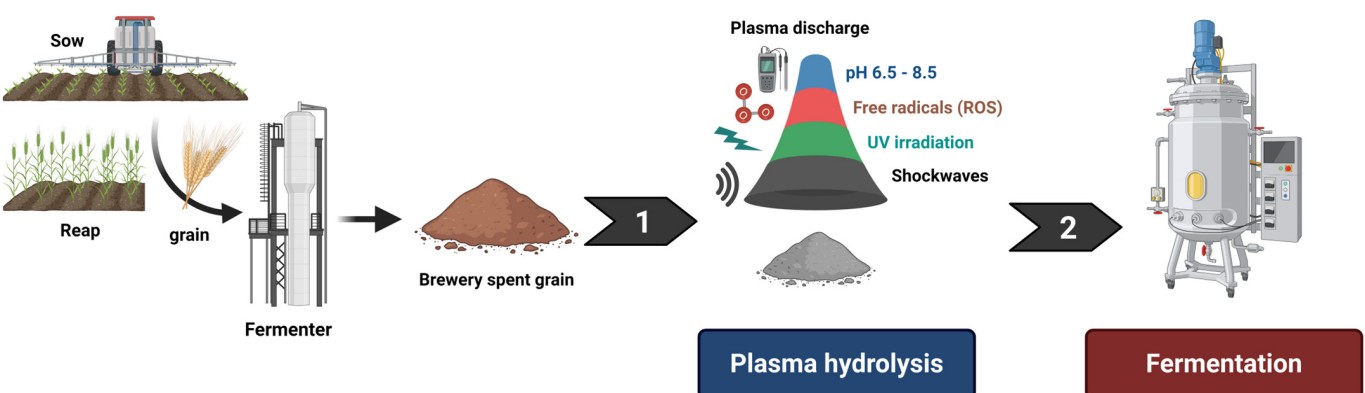

**Figure 3.** Schema for incorporating plasma hydrolysis of BSG to enhance enzymatic digestion and subsequent fermentation pathways. Adapted from [24]. Created in BioRender.

## 3. Processing Methods

The production of PHAs starting from brewery spent grain (BSG) can be based on different production methods, such as mixed microbial cultures (MMCs), aimed at the production of volatile fatty acids (VFAs), and solid-state fermentation (SSF) techniques [6]. MMCs have been extensively explored for their ability to produce PHAs efficiently [25], while SSF has emerged as a cost-effective strategy for PHA synthesis [26].

### 3.1. Mixed Microbial Cultures (MMCs)

Polyhydroxyalkanoate (PHA) production from brewery spent grain (BSG), utilizing mixed microbial cultures (MMCs), involves a series of key steps. Initially, the selection of MMCs enriched in efficient PHA-accumulating organisms is crucial for the success of the process [1]. The enrichment of MMCs in PHA-storing microorganisms is then carried out to establish a robust system for PHA production from BSG [27]. BSG, a byproduct of the brewing industry, efficiently serves as a carbon source for PHA production by MMCs, contributing to the overall sustainability of the process [1]. PHA accumulation occurs within the MMCs under conditions of transient carbon availability, necessitating dynamic feeding strategies to optimize PHA storage [28]. Process optimization is continuously pursued to enhance PHA production using MMCs and waste streams as carbon sources, aiming to reduce costs and improve upcycled efficiency [29]. Following PHA accumulation, the extracted PHAs undergo characterization to evaluate their properties, including an analysis of polymer characteristics to ensure the desired material properties are achieved [30].

In the realm of PHA production from BSG using MMCs, these steps are crucial for the successful and sustainable generation of bioplastics [30]. By navigating a pipeline in which MMCs are first selectively enriched for PHA-producing phenotypes on a nutritional fermentation palette of reclaimed BSGs as carbon sources, researchers can fine-tune the optimal production process while characterizing the synthesized PHAs for subsequent bioplastic manufacturing methods [27–30]. With the promise of future refinement to optimize the BSG-PHA production pipeline, researchers and industry practitioners can thus advance the utilization of waste materials for environmentally friendly bioplastic production (Figure 4).

Biologically, the MMC approach revolves around volatile fatty acids (VFAs), which serve as the precursors for PHAs and are typically derived from organic matter [6,26]. The utilization of VFAs obtained through acid fermentation of organic substrates by MMCs presents economic advantages, such as low production costs associated with open systems and cost-effective substrates. These factors contribute to the potential for reducing overall PHA production costs and enhancing the at-scale economic feasibility of PHA production using MMCs.

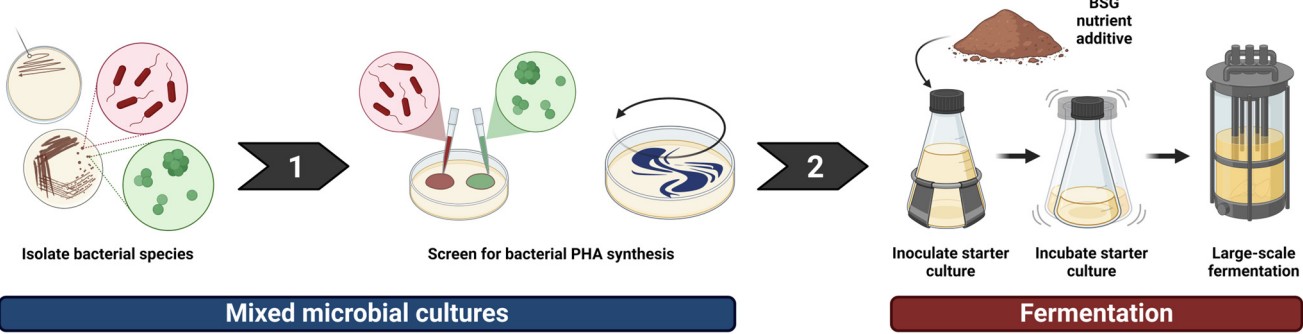

**Figure 4.** Schematic representation of PHA production via mixed microbial enrichment cultures without pretreatment methodologies. Modified from [31]. Created in BioRender.

Despite promising economic benefits, challenges persist in the production of PHAs through MMCs. One significant hurdle involves reproducibly synthesizing uniform PHA composition, which is crucial for ensuring consistent biomaterial properties and quality of the final bioplastic product. Variability in PHA composition can greatly impact the material's performance and applicability in various industries. Additionally, optimizing productivity from waste materials poses a challenge, as the efficiency of PHA production from organic substrates by MMCs may vary depending on the precise chemical composition and availability of the feedstock. This will likely entail a precise microbiome characterization custom tailored to each nutritional input, as different PHA-producing microbial species favor differing feedstock during fermentation pathways. Defining such microbiomes will greatly pave the way for optimizing a reliable and efficient bioreactor process for converting waste materials into PHAs as a crux for sustainable production practices.

Furthermore, maintaining cost competitiveness compared to conventional PHA production methods is another challenge faced by MMC-based PHA production. While the use of MMCs and waste substrates offers potential cost advantages, the scalability and efficiency of the process will require optimization to compete with traditional PHA production methods in terms of cost-effectiveness and production volume. Addressing these challenges through research and development efforts focused on process optimization, microbial engineering, and feedstock selection is essential for advancing MMC-based PHA production and enhancing its competitiveness in the bioplastics market.

### 3.2. Solid-State Fermentation (SSF)

Polyhydroxyalkanoate (PHA) production from brewery spent grain (BSG) using solid-state fermentation (SSF) involves several key steps. Initially, BSG is utilized as the substrate for PHA production through SSF [32]. The lignocellulosic structure of BSG is hydrolyzed to make the cellulose more accessible to enzymes, followed by enzymatic hydrolysis to obtain a saccharified solution containing glucose as the main sugar [33]. Subsequently, the hydrolysate undergoes fermentation by suitable microorganisms, such as *Lactobacillus* spp., to produce a variety of PHAs [33]. This process highlights the importance of pretreatment steps enacted on raw BSG inputs to enhance the nutritional accessibility of substrates for microbial metabolism and fermentation for PHA production.

In SSF, the solid-state nature of the fermentation process presents additional challenges due to the recalcitrant and lignocellulosic content of BSG [1]. Pretreatment steps are often needed to improve the process performance, as the solid-state form of BSG requires specific conditions for efficient biological breakdown processes [1]. However, the use of SSF allows for the sustainable production of PHAs from BSG, showcasing the versatility of this fermentation method in valorizing waste materials [1].

Furthermore, the SSF of BSG involves the production of various value-added compounds, including organic acids, amino acids, enzymes, vitamins, and second-generation biofuels [34]. The optimization of SSF conditions, such as solid loadings and fermentation parameters, is essential to achieve high yields of target products like PHAs [34]. The application of SSF for PHA production from BSG demonstrates the potential for utilizing this waste material in biotechnological processes to generate valuable bioproducts in a sustainable fashion [34].

Advancements in PHA production from residue-based systems predominantly utilize submerged fermentation methods, encompassing diverse source materials from municipal wastewater to sugarcane molasses [9]. The study by [9] introduces SSF as a distinctive approach, emphasizing microbial growth on moist solid particles [9,18]. Notably, *Burkholderia cepacia* demonstrates superior poly-3-hydroxybutyrate (PHB) accumulation using glycerol and molasses under SSF, showcasing advantages such as lower energy needs, enhanced cost savings, and elevated efficacy compared to submerged fermentation [9].

The investigation further explores breweries as potential contributors to a bio-based circular economy, operating globally, year-round. Brewer's spent grain (BSG), constituting 85% of brewing byproducts, are examined for their biomass fractionation potential to produce valuable chemicals and biomaterials, aligning with the pyramid of bio-based products to prioritize higher-value byproducts [35].

Solid-state fermentation (SSF) for obtaining polyhydroxyalkanoates (PHAs) from beer spent grain can be either aerobic or anaerobic, depending on the specific microbial strains used and the desired metabolic pathways for PHA production.

Aerobic solid-state fermentation involves the presence of oxygen during the fermentation process, which can be beneficial for certain microorganisms that require oxygen for their metabolic activities. In aerobic conditions, microorganisms can efficiently convert substrates into desired products like PHAs. On the other hand, anaerobic solid-state fermentation occurs in the absence of oxygen and can be suitable for microorganisms that thrive in oxygen-deprived environments. Anaerobic fermentation can also lead to the production of PHAs under specific conditions and has been demonstrated in the valorization of various waste streams including municipal solid waste [16,36].

The choice of microorganisms for solid-state fermentation in PHA production from beer spent grain is crucial. Various microbial strains can be utilized for PHA production, including but not limited to bacteria belonging to the genera *Bacillus*, *Burkholderia*, and *Cupriavidus* [2]. These microorganisms have the capability to accumulate PHAs efficiently from substrates like BSG under suitable fermentation conditions. Selection of the appropriate microbial strains is thus essential to ensure high PHA yields and efficient conversion of the substrate into biodegradable polymers.

## 4. Results

Within the context of PHA production using mixed microbial cultures (MMCs), the enrichment process driven by escalating organic loading rates (OLRs) to selectively accumulate PHA-storing microorganisms is well described in the findings of [1]. The enrichment process, driven by escalating organic loading rates (OLRs), showcases the selective accumulation of PHA-storing microorganisms, notably operational taxon units drawn from the *Meganema*, *Carnobacterium*, *Leucobacter*, and *Paracoccus* genera. The specialized microbiome attains a commendable PHA content of approximately 35% assessed via volatile suspended solid assays [1]. The MMC permutation comprising *Meganema*, *Carnobacterium*, *Leucobacter*, and *Paracoccus* genera reflects a highly specialized microbiome suited for BSG conversion to PHA [1]. However, outside of BSG valorization, an extensive body of research exists for identifying precise bacterial organisms with the aim of charting each organism's capacity for PHA production across a wide range of nutrient inputs [37,38].

The feasibility of a sustainable biorefinery system for PHA production from lignocellulosic-derived wastes, including BSG, was also examined [39]. Enzymatic cocktails for BSG degradation can be induced by fungal microbes, *Aspergillus brasiliensis* and *Trichoderma reesei*, during SSF protocols. In turn, these induced enzymatic cocktail extracts, which likely comprise a cadre of ligninases and cellulases, are quite effective alongside ionic liquid exposure in releasing fermentable sugars during breakdown of BSG biomass. [39]. The coupling of SSEH and SSF approaches in turn amplifies PHA yields by up to 54% for BSG, in comparison to SSF alone [18].

Moreover, the utilization of BSG as a carbon source for the fermentation production of biodegradable polyesters (i.e., PHAs) and the successful synthesis of PHB and PHB-co-MCL from BSG hydrolysates align with the findings in [10,40]. The achieved biopolymer titers in the study, PHB at 3.53 g/L and PHB-co-MCL at 3.32 g/L, substantiate the potential of BSG as a raw material for PHA production, thereby reducing production costs through the utilization of valued waste resources [10,40]. Thus, much progress has been made to underscore the feasibility and overall sustainability of PHA production from fermented BSG and agricultural wastes, emphasizing the valorization of waste streams and concomitant development of biodegradable products for various applications [1,10,39,40].

## 5. Discussion

The utilization of brewery spent grain (BSG) for the production of biopolymers emphasizes its significant potential in sustainable and cost-effective valorization endeavors (Figure 5). This approach aligns seamlessly with the principles of a circular economy by transforming a byproduct into valuable materials. Particularly, the production of polyhydroxyalkanoates (PHAs) and polyhydroxybutyrates (PHBs) has demonstrated notable economic advantages, with BSG serving as an abundant and low-cost substrate [10,14,41].

The versatility of BSG is evident through its application in various biopolymers, including PHAs and PHBs. The continued exploration of alternative polymers like polyhydroxybutyrate-co-valerate (PHBV) and polylactic acid (PLA) from BSG shows burgeoning promise with detailed specifics drawing from several interdisciplinary approaches [15,42,43]. Thus, continued research is imperative to enhance efficiency and refine the cost-effectiveness of these emerging processes to drive progress towards a fully integrated circular economy.

### 5.1. Pretreatment Processes

BSG is a lignocellulose biomass material comprising a highly complex matrix of cellulose, hemicellulose, and lignin molecules. Due to the highly recalcitrant nature of the lignocellulose biomass, the BSG needs to undergo different pretreatment processes for efficient sugar extraction using purified enzymes or enzyme cocktails derived from fungal growth.

In the hydrothermal process or diluted acid hydrothermal process, the biomass is heated under high temperatures, typically 160 to 220 °C, under steam sterilization. At this temperature point, both hemicellulose and lignin become solubilized and undergo a hydrolysis process, where a portion of hemicellulose is converted to acid to further assist the hemicellulose hydrolysis process [45]. However, the hydrolysis of lignin and cellulose inadvertently causes byproduct metabolites that are known to inhibit downstream bacterial fermentation pathways [46,47]. An alternative approach is to use concentrated acid pretreatment under a high operating temperature to achieve efficient hydrolysis [48]. However, this requires the chemical reactors or vessels to be highly resistant to corrosion. The recovery of acid and neutralization poses a significant challenge for downstream processes and waste disposal. Similar issues are also present in its strong alkaline approach.

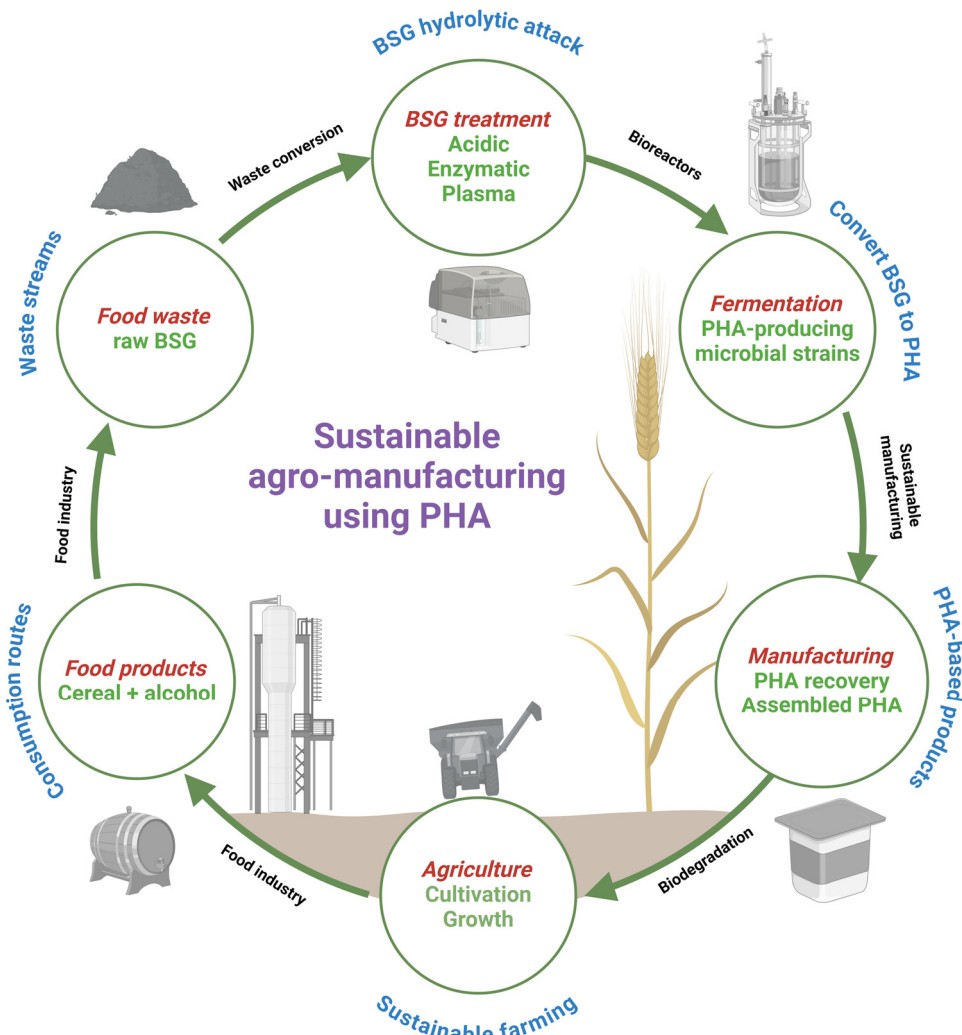

**Figure 5.** Circular economic schema for sustainably tethered agricultural and manufacturing methodologies leveraging PHA. Modified from [44]. Created in BioRender.

Encouraging progress has also been made in evaluating nonthermal pretreatment processes for releasing fermentable sugars from BSG waste streams. Ravindran and others proposed the use of discharge plasma to reduce recalcitrance and delignify the BSG [24]. The group reported a high reducing sugar yield for bioethanol production. Despite being under mild temperature, the discharge plasma required a high electrical voltage of up to 28 kV, posing additional safety concerns [24].

The combination of different pretreatments is expected to improve the sugar yield with reducing inhibitors. Ponsá and others compared different pretreatment conditions [2], with thermal acid treatment providing higher sugar concentrations. Other methods, such as the combination of a nonthermal microwave–alkaline approach also causes a high sugar yield, which is only slightly lower than the acid–thermal process [2]. However, the PHB production yield from this approach is lower due to the presence of a high number of phenolic compounds because of alkaline pretreatment [2]. The same group also reported adding a wash step after thermal acid treatment to remove inhibitors [2]. It is promising to note that this approach achieved an enhanced PHB production yield with the highest observed sugar conversion rates at approximately 57 mg PHB per gram of reducing sugar, while raw BSG (i.e., lacking any form of pretreatment) alone yielded roughly 44 mg of PHB g$^{-1}$ reducing sugar using the bacterial organism, *Burkholderia cepacia* [2]. These findings indicated that the production yield of PHAs in liquid cultures may not need a high initial sugar concentration for further fermentation. It is expected that future development

can be focused on obtaining an optimized order of pretreatment to release significant fermentable sugars, while lowering the inhibitors.

### 5.2. Temperature Effects on Microorganisms

In the production process, pH and temperature are critical factors that significantly influence the synthesis of volatile fatty acids (VFAs), essential precursors in bioplastic production. Research primarily focuses on mesophilic conditions—those in the moderate temperature range of 20 °C to 45 °C (68 °F to 113 °F)—which is amenable for the laboratory evaluation of bacterial fermentation processes. This is because mesophilic microorganisms thrive at these temperatures, making it a standard choice for efficient VFA production without the need for excessive heating or cooling. Nonetheless, even amongst mesophiles, *Cupriavidus necator* was a better PHA producer than *Burkholderia cepacia* during operational fermentation temperatures at 30 °C on hydrolyzed BSG nutritional inputs [32].

However, researchers have also extended exploratory investigations into psychrophilic (cold-loving) and thermophilic (heat-loving) microbial taxa to uncover potential energy savings. Psychrophilic organisms operate effectively at temperatures below 20 °C (68 °F), offering a unique advantage in reducing the energy required for heating the system, especially in climates or seasons where ambient temperatures are lower than desired for the process. This could lead to significant energy savings in maintaining the production environment, particularly in reducing the reliance on artificial heating. Halophilic and psychrophilic bacteria, such as *Halomonas* spp. and *Paracoccus* spp., are advantageous for PHA production due to their high salinity tolerance and growth at low temperatures, reducing contamination risks and production costs [49].

On the other hand, thermophilic organisms flourish at temperatures above 45 °C (113 °F). While initially it might seem counterintuitive to seek energy savings in processes requiring higher temperatures, thermophilic conditions can accelerate reaction rates and microbial activity, potentially reducing the process time and improving efficiency. Moreover, in some setups, the process might utilize residual heat from other industrial operations or even generate enough heat internally to maintain these higher operational temperatures, thereby reducing the need for external heating sources. A thermotolerant microorganism, *Pseudomonas* sp. strain SG4502, was successfully isolated and empirically shown to be capable of accumulating medium-chain-length polyhydroxyalkanoate (mcl-PHA) from a BSG byproduct as a carbon source at a cultivation temperature of 45 °C [50].

The aim of exploring these temperature ranges—mesophilic, psychrophilic, and thermophilic—is to identify the most energy-efficient fermentation processes for VFA production. By understanding and leveraging the unique characteristics and environmental preferences of different microorganisms, researchers and industry practitioners can optimize bioplastic production processes, minimizing energy consumption while maximizing yield and efficiency.

### 5.3. Doehlert Design

The Doehlert design [51] optimizes operating parameters for VFA production from raw and processed brewery spent grain, revealing correlations among parameters, substrate, VFA profiles, and bacterial populations.

Microbial metabolic pathways that yield VFAs are inherently intricate processes, and an understanding of optimal pathway control is critical in the production of sustainable bioplastics. The Doehlert design approaches this complexity not as a maze of independent variables but as a dynamic ecosystem where each element—pH, microbial concentration, and temperature—plays a pivotal role. By applying this design, one can systematically vary these parameters across a strategically defined experimental space, revealing their individual and collective impacts on VFA production.

For instance, the optimization process might begin by setting a series of experiments where pH levels vary alongside microbial concentration and temperature. The design ensures that each variable is adjusted in a coordinated manner, creating a comprehensive map of the experimental landscape. As a further example of a refined Doehlert design, numerous, controlled parameters can be evaluated in an experiment where pH is adjusted from slightly acidic to neutral, microbial concentration is varied to include both low and high densities, and temperature is modulated within a strict mesophilic range.

Through the lens of the Doehlert design, this approach reveals not just the direct effects of each variable on VFA yield but also how these variables interact as a systems level. Such an approach might uncover how optimal VFA production occurs at a specific pH when microbial concentration and temperature are at their mid-range values. Conversely, the design could demonstrate how elevated temperatures amplify the effect of pH adjustments or how microbial concentrations become a limiting factor under certain temperature and pH conditions. These are all important considerations when arriving at a sustainable solution for the valorization of BSG into biodegradable bioplastics, as concerns a circular economy.

### 5.4. Economic Analysis and Optimization of BSG Utilization

The adoption of BSG as a substrate for PHA production necessitates crucial economic evaluation and strategic optimization. While BSG offers cost advantages as a low-cost or no-cost raw material, potentially reducing the substantial 30–50% raw material costs typically associated with biopolymer production [10,52], these benefits are offset by the need for its pretreatment to render it suitable for microbial fermentation. This pretreatment may add operational complexities and financial burdens, including capital, operational, and chemical costs. However, the cost of pretreatment could be mitigated by scaling up and optimizing the process's flexibility [53].

Moreover, the yields of PHAs from BSG typically do not match those from refined substates like glucose and sucrose; for example, PHA yield from glucose and sucrose in single-culture conditions are reported as 300 g/kg and 170 g/kg, respectively [54], compared to 12.5 g/kg dry added residue [18], 13.1 g/kg via SSF [9], and 90 g/kg pretreated BSG [10]. This lower yield could lead to higher operational costs per unit of PHA produced, as more substrate is required to achieve the same output.

Despite these challenges, the environmental benefits of using BSG are significant. Utilizing BSG supports the principles of a circular economy by reducing waste and promoting the valorization of byproducts. The environmental appeal of this approach could also enhance the marketability of the resulting PHAs, potentially allowing for premium pricing based on the sustainable nature of the production process and offsetting some of the costs associated with yield inefficiency and pretreatment processes. Therefore, ongoing research and development, supported by suitable policy incentives, are imperative to refine the economic and technical viability of BSG in commercial PHA production settings.

## 6. Conclusions

The integration of BSG into biopolymer production processes is a sustainable and economically viable approach (Figure 5). The adoption of efficient fermentation methods, optimization of pretreatment and hydrolysis processes (Figures 1–3), and the exploration of diverse microbial cultures (Figure 4) significantly contribute to the overall feasibility of BSG-based biopolymer production. Adopting a circular economy approach (Figure 5) while simultaneously mindful of cost considerations, traditional waste streams including brewery spent grain (BSG) is instead upcycled into an asset for producing environmentally friendly, biodegradable polymers. This perspective promotes sustainable agricultural/industrial methodologies by emphasizing waste repurposing and resource preservation (Figure 5).

**Author Contributions:** Conceptualization, D.P. and H.D.K.; methodology, R.D.T. and P.N.P.; formal analysis, M.D.G. and G.V.; investigation, R.D.T., P.N.P., D.P. and M.D.G.; resources, H.D.K., X.J. and L.C.; data curation, P.N.P. and R.D.T.; writing—original draft preparation, R.D.T., P.N.P. and D.P.; writing—review and editing, D.P., M.D.G., G.V., H.D.K. and L.C.; visualization, G.V. and M.D.G.; supervision, D.P., M.D.G., X.J. and L.C.; project administration, D.P., M.D.G. and X.J.; funding acquisition, D.P., G.V., M.D.G. and X.J. All authors have read and agreed to the published version of the manuscript.

**Funding:** This research was funded by the Manufacturing PA Innovation Program, grant number 1060170-457922.

**Data Availability Statement:** No new data were created in this study. Data sharing is not applicable to this review article.

**Acknowledgments:** We wish to express gratitude to both the engineering and biology departments of Gannon University for nurturing and sustaining deep interdisciplinary collaborations.

**Conflicts of Interest:** The authors declare no conflict of interest.

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
