# Peer review of "Harnessing Brewery Spent Grain for Polyhydroxyalkanoate Production"

_2673-6209, doi:10.3390/macromol4030026_

Round 1
Reviewer 1 Report
Comments and Suggestions for Authors
The report presents a review of using Brewery Spent Grain as the ecologic resource of some biopolymers polyhydroxyalkanoates family. It does not introduce essential new results, but such a comprehensive presentation with references can be interesting for some readers. It is a pity that the authors did not attempt to formulate the research and application 'added value' resulting from such a comparison, which would have significantly raised the rank of the work. In any case, it is not explicitly visible. Perhaps a table summarizing research and application results, as well as containing an economic reference for the applications, would be helpful to the reader. The latter is implicitly 'promised' in the introductory parts of the work but is not satisfactorily emphasized. Such a table, perhaps supported by an 'intelligent diagram', could constitute a simple, initial orientation diagram for the methodology and applications.Koniecznie należy poprawić jakość i czutelność Figures 1, 2, and 3.
Reviewer 2 Report
Comments and Suggestions for Authors
1. The figures in this manuscript are not clear, due to low resolution.
2. The conclusion section need be rewrite, and authors should point out the problems of polyhydroxyalkanoates production from brewery spent grain.
3. Authors can draw a graphical abstract for this review.
4. The discussion section is not enough. Please discuss more factors for polyhydroxyalkanoates production using more references.
Comments on the Quality of English LanguageAuthors need revise English writing errors and correct punctuation marks, for example, Consider the intricate process of volatile fatty acid (VFA) synthesis, a critical pathway in the production of bioplastics, the Doehlert design approaches this complexity not as a maze of independent variables but as a dynamic ecosystem where each element—pH, microbial concentration, and temperature—plays a pivotal role.
Round 2
Reviewer 2 Report
Comments and Suggestions for Authors
No.
Comments on the Quality of English LanguageNo.